# School counselor advocacy for gender minority students

**Jack D. Simons**  *, on behalf of the Simons Lab[¶]

Counseling, Mercy College, Dobbs Ferry, New York, United States of America

¶ Membership of the Simons Lab is listed in the Acknowledgments.
* jsimons1@mercy.edu

## Abstract

One-thousand-one-hundred-and-ninety-one school counselors completed an online survey regarding advocacy for and with gender minority students comprising transgender and intersex students (school counselor gender minority advocacy competence). School counselors completed a battery of three competency-based assessments to assess their levels of gender identity counselor competence, intersex counselor competence, and school counselor gender minority advocacy competence. They also completed a demographic form so that competency levels could be examined with demographic variables. Attitudes, school level placement, gender, sexual orientation, gender identity counselor competence, and intersex counselor competence were significantly related to advocacy for and with gender minority students. These findings have implications for the application of identity behavior theory to examine the experiences and behaviors of minoritized individuals and those who advocate for them. These advocates include school counselors and other helping professionals who work with gender minority students.

## Introduction

A paucity of research exists on school counselor gender minority advocacy competence, the extent to which school counselors effectively advocate for and with transgender and intersex students [1]. This lack of research is concerning because gender minority youth are becoming more visible in society, and, although the American School Counselor Association has called for respect and equal treatment of these students, many still report negative school experiences [2, 3]. Gender minority youth attending unsafe schools report being bullied, harassed, and victimized [2, 3]. Concurrent with major changes in society concerning attitudes toward gender minorities, and a need for more scholarship in this area, the adults who are expected to support them are encouraged to look more closely at their ability to advocate for gender minorities [1].

In this study gender minority youth comprise those who identify as transgender and intersex. Scholars estimate 150,000 adolescents in the United States identify as transgender [4]. Transgender youth experience incongruent feelings between birth sex and gender identity [5]. Intersex individuals account for one to two percent of the population [6]. Intersex youth possess a normal variation in hormone levels or chromosomes and may experience differences in body characteristics [7]. Some will undergo genital surgery [8]. Some transgender and intersex

**Data Availability Statement:** The data underlying this study is available in the openICPSR data repository (https://doi.org/10.3886/E122501V1).

**Funding:** Research reported in this article is part of a Faculty Development Grant supported by Mercy

College under IRB protocol number 17-75. The College funders did not a play a role in the study design, data collection and analysis, decision to publish, and preparation of the manuscript.

**Competing interests:** The authors have declared that no competing interests exist.

youth also identify as gender non-binary. Gender non-binary youth, also referred to as gender-queer youth, possess non-dichotomous gender identities that are neither male nor female; the identities of gender non-binary youth may be fluid or fixed, or exist somewhere between female and male (e.g., neutral) [9, 10].

Examining counselor advocacy for transgender and intersex youth is warranted because the needs of these youth appear to overlap [2, 3]. The two groups experience a wide array of feelings; face challenges (e.g., being bullied); and cope with identity development [2, 3]. This is also the case for gender non-binary students. As a result, the aim of this quantitative study is to assess the school counselor role as an advocate for transgender and intersex students. However, more research is needed to understand counselor advocacy for gender non-binary youth. The decision was made not to include assessment of advocacy for these students because prior to this study, the research team had only conducted studies to examine the school experiences of transgender and intersex students, not the school experiences of gender non-binary students [2, 3]. Additionally, school counselor gender minority advocacy competence comprising transgender and intersex advocacy competence (together) was examined to develop and norm the School Counselor Transgender Intersex Advocacy Scale, an online self-administered training tool that is used by school counselors and other school stakeholders to examine their ability to advocate for gender minority students [11]. This scale is an important tool because the amount of time that school counselors have for training may be limited due to high caseloads and other work responsibilities. By combining gender minority related training content together (transgender and intersex), school counselors may learn about how effective they are at meeting the needs of these students in a shorter amount of time. A review of the current body of research suggests the following regarding the need for more effective training of school counselors and other helping professionals who work with gender minority youth:

1. We do not know how counselor competence to provide services to transgender students relates to counselor competence to provide services to intersex students. A review of the research literature indicates that no studies have empirically examined these advocacy areas together. To the best of our knowledge, this study appears to be the first.

2. We have limited knowledge about how to train school counselors to most effectively support gender minority students, but what we do know, most notably from Australia, informs the research area [12–16]. Jones et al. [12] assessed mixed research data to learn more about the experiences and identities of both transgender and intersex students in Australia. The sample consisted of 189 gender diverse students who were 14 to 25 years of age. All were surveyed and 16 participated in qualitative interviews. Findings indicated that the needs of the participants varied widely, and their identities were not fixed. Transgender students reported that they did not display gender(s) congruent with the expectations of others. The students, therefore, valued learning about human development. As a result, it appears administrators and counselor educators should prioritize teaching current and future school counselors about how to discuss these areas with students and other school stakeholders, including parents. McGuire et al. [15] analyzed mixed data from 67 transgender students. The students reported that it was common for them to experience harassment and feel unsafe unless they were enrolled in schools with gender-inclusive curricula. As compared to cisgender students, they were less likely to have adequate academic and familial support, and some thought of suicide. Riggs et al. [16] analyzed survey data retrieved from 28 school stakeholders comprising cisgender school counselors, psychologists, and parents. The findings suggested that counselors were uniquely situated in schools to advocate for transgender students but needed more education to effectively do so [16]. Findings from the studies of McGuire et al. [15] and Riggs et al. [16] suggest that school counselors

should be taught about how transgender students benefit from having teachers who have knowledge of transgender issues such as social and familial support, mental health, and inclusivity. For example, regarding the latter, school counselors should call for and run gender minority and ally groups and promote policies to assist gender minority students to transfer schools if needed (e.g., due to ongoing physical assault) [2].

3. We do not know how counselor competence to provide counseling services for and with transgender and intersex students within a comprehensive model of school counseling relates to counselor competence to provide counseling services to transgender and intersex individuals in a traditional model of one-on-one individual counseling. For example, with even controlling for gender and religious affiliation, would those who provide comprehensive model school counseling to the parents of gender minority youth be as effective as those who provide traditional individual counseling services to the parents of gender minority youth? [16].

## Competency-based assessment

Training school counselors to self-reflect over who they are and how it influences the services they provide may be facilitated by use of competency-based assessments (CBAs). According to Lurie [17], CBAs illustrate models of competency that can be developed and refined by individuals. Educators in the health professions (e.g., counselors) and those who train them (e.g., counselor educators) use CBAs to gather empirical data to examine and improve best practices. Use of CBAs remain widespread despite the belief by some that CBAs overemphasize the development of individual skills to the detriment of not learning how to remain open to gaining knowledge from the totality of one's learning experience (holism). Brightwell and Grant [18] have argued that this inability to remain open to totality of experience weakens the role of trainees and hurts professions. One way to address these concerns might be to emphasize to students why making a commitment to holistic lifelong learning early in their careers is important [18]. More research, however, is warranted to develop more holistic and reflexive approaches to using CBAs in training and research. According to Bajis, Chaar, and Moles [19], the inclusion of lifelong learning into CBAs recognizes that competencies change over time due to ethical, social, clinical, and technological considerations. Addressing lifelong learning in CBAs is newer; highlighting the need for lifelong learning in CBAs has the ubiquitous goal of advancing competency paradigms over the entire lifespan in an ever-changing, dynamic world [19].

CBAs are used as part of educational practices, including accreditation, and to globalize behavioral expectations (e.g., advocating for gender minority students throughout the world). Individuals who effectively develop and modify CBAs to use in training consider historical, current, and future trends as well as try to strike a balance between use of prescriptive and descriptive language. To norm CBAs, Lurie [17] recommended framing constructs in terms of data-based hypotheses, multiple situations, and reliability so that results from CBAs can be compared. Researchers and counselor educators have used CBAs to teach about multicultural counseling, group counseling, and internship training, to name just a few. More recently, counselor competencies concerning gender have been identified, assessed, and taught using CBAs [11].

## Identity behavior theory

Simons [20] proposed Identity Behavior Theory as an alternative to Planned Behavior Theory (PBT). PBT has been widely used to predict behavior resulting from others' expectations, one's

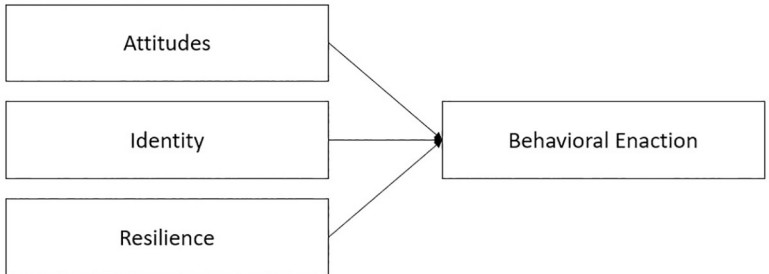

**Fig 1. Identity behavior theory.** Conceptual representation of the elements of Identity Behavior Theory described by Simons.

attitudes, self-efficacy, and behavior plans (intention) [20]. Behaviors examined using PBT have included, but are not limited to, health and food choices, educator effectiveness, and knowledge sharing; however, despite being developed over three decades ago, few changes have been made to PBT to improve its predictive validity. Moreover, the model does not explicitly recognize the role that identity plays in influencing behavior, nor does it emphasize the emancipatory nature of people, particularly sexual and gender minorities and their allies [20]. Identity Behavior Theory (IBT), in contrast to PBT, posits that individuals enact behavior based upon their attitudes, resilience, and identities (see Fig 1). Identity and resilience (personal strength and support) serve as a foundation for behavioral enaction (preparing to act, trying to act, and successfully acting).

In this study, we examine the underpinnings of school counselor gender minority advocacy in light of IBT using three CBAs: the School Counselor Transgender Intersex Advocacy Competence Scale, the Gender Identity Counselor Competence Scale, and the Intersex Counselor Competence Scale [11]. Each of these measures will be further discussed and are comprised of subscale measures that assess attitudes, resilience, and enaction. Additionally, we examine how school counselor advocacy for gender minority students is related to several demographic variable (see S1 File). The research questions are:

- Is gender identity counselor competence related to school counselor gender minority advocacy competence?

- Is intersex counselor competence related to school counselor gender minority advocacy competence?

- Are school counselor characteristics (age, attitudes, race/ethnicity, gender, sexual orientation, and school level placement) related to school counselor gender minority advocacy competence?

Given the tenets of CBAs and IBT, we hypothesized that school counselor gender minority advocacy competence would relate to gender identity counselor competence, intersex counselor competence, and school counselor characteristics (age, attitudes, race/ethnicity, gender, sexual orientation, and school level placement). In addition, we believed that findings would show: (a) a significant positive relationship between intersex counselor competence and school counselor gender minority competence; (b) a significant positive relationship between gender identity counselor competence and school counselor gender minority advocacy competence; and (c) significant differences between school counselors' characteristics (age, attitudes, race/ethnicity, gender, sexual orientation, and school level) on school counselor gender minority advocacy competence.

## Methods

The Mercy College IRB approved the study with an approval number of 17–75. Study participants gave consent to participate via electronic REDCap survey form. A national survey study was conducted with a sample of school counselors. A nonprobability sample was sought to increase the likelihood of response. Participants were not exposed to any more risk than an average person in the general population would be exposed to when going about daily activities. We recruited school counselors throughout the United States by distributing a study announcement using email addresses, social media platforms, and contacts at state certification departments and state counseling associations. To learn more about recruitment procedures and geographical data, refer to Simons [11] and Simons, Bahr, and Ramdas [21].

The three CBAs and a demographic form were self-administered by 1,191 school counselors during the 2018–2019 school year. Data were gathered from school counselors and then downloaded from REDCap survey platform. Thereafter, data were cleaned, and reliable simulated values were created to impute missing data. All scale items had some missing data except for the demographic items. Forty-five percent of participants had at least 1 value missing. Eight percent of values had missing data. Five percent were missing completely at random, and three percent were missing based on sexual orientation. This latter missing data were on intersex topics. This suggests that one's sexual orientation might be related to attitudes, knowledge, and skills concerning the intersex community. This might also suggest that participants had more knowledge and exposure to transgender topics than intersex topics. The items with the greatest amount of non-response were the following two items: (a) Prejudicial concepts and intersexism (i.e., oppression towards intersex people) have permeated the mental health professions, and (b) Personally, I think being intersex is a mental disorder or a sin and can be treated through counseling or spiritual help. The average mean was used to replace all missing data.

## Instruments

The demographic form included items on age, race/ethnicity, gender, sexual orientation, and school level. It was completed along with the School Counselor Transgender Intersex Advocacy Competence Scale (SCTIACS), the Intersex Counselor Competence Scale (ICCS), and the Gender Identity Counselor Competence Scale (GICCS). The SCTIACS is a 70-item scale comprising Likert-type scale items: a 22-item attitudes subscale and a 48-item advocacy subscale [11]. Items on the scales range from 1 (not at all true) to 6 (totally true). The SCTIACS is a CBA used to self-assess the degree to which one effectively advocates for gender minority students based upon attitudes, resilience, and identity. This is referred to as school counselor gender minority advocacy competence. Sample items include: (a) I have reflected over my own attitudes, beliefs, and transgender and intersex advocacy practices, (b) I have knowledge of the transgender identity developmental models, and (c) I am knowledgeable of the American School Counselor Association's (2016) position statement on counseling transgender and gender non-conforming students [22]. Scores on the SCTIACS range from 70 to 420. School counselors who scored above 245 were deemed to have had high levels of gender minority advocacy competence, and school counselors who scored below were deemed to have had low levels of gender minority advocacy competence. The attitudes subscale, the Awareness of Transgender and Intersex People Scale (ATIPS), assesses how one views gender minority individuals. Scores on the ATIPS range from 22 to 132. School counselors who scored above 77 were deemed to have held more positive attitudes toward gender minority individuals, and school counselors who scored below 77 were deemed to have held fewer positive attitudes toward gender minority individuals. The advocacy subscale, the Transgender Intersex

Advocacy Activity Scale (TIAAS), assesses school counselors' knowledge and skills. Scores on the TIAAS range from 48 to 288. School counselors who scored above 168 were deemed to have held more knowledge and skills about advocating for gender minority individuals, and school counselors who scored below 168 were deemed to have held less knowledge and skills about advocating for gender minority individuals. Examples of knowledge assessed included knowledge of stereotypes, the coming out process, transgender identity developmental, and how to assist with gender transition. Examples of skills assessed include promoting anti-bullying policies, providing training and awareness programs, consulting with parents and guardians, and evaluating the strengths and weaknesses of school counseling programs.

The Intersex Counselor Competence Scale (ICCS), another CBA, is a 27-item scale comprising Likert-type scale items: a 9-item skills subscale (ICCS-S) and a 7-item knowledge and beliefs scale (ICCS-KB). Items on the scales range from 1 (not at all true) to 7 (totally true). The ICCS measures the self-perceived competence of one concerning ability to provide individual counseling services to individuals who identify as intersex [23]. Sample items include: (a) I have experience counseling female-to-male transgender clients, (b) I have been to in-services, conference sessions, or workshops, which focused on transgender issues in counseling, and (c) There are different psychological/social issues impacting transgender men versus transgender women. Scores on the ICCS range from 16 to 112. School counselors who scored above 64 were deemed to be more effective as counselors working with intersex individuals, and school counselors who scored below 64 were deemed to be less effective at counselors working with intersex individuals. The ICCS-S assesses the skills of counselors who work with intersex individuals. Scores range from 9 to 63. School counselors who scored above 36 were deemed to have possessed more skill at counseling intersex individuals, and school counselors who scored below 36 were deemed to have possessed less skill at counseling intersex individuals. The ICCS-KB assesses the knowledge and beliefs of counselors who work with individuals who identify as intersex. Scores on the ICCS-S range from 7 to 49. School counselors who scored above 28 were deemed to have possessed more knowledge and favorable beliefs about counseling these individuals, and school counselors who scored below 28 were deemed to have possessed less knowledge and favorable beliefs about counseling these individuals.

The Gender Identity Counselor Competence Scale (GICCS) is a 29-item CBA that measures the self-perceived competence of one's ability to provide individual counseling services to individuals who identify as transgender [24]. Likert-type scale items range from 1 (not at all true) to 7 (totally true). Sample items include: (a) Currently, I do not have the skills or training to do a case presentation or consultation if my client was intersex (b) I feel that gender differences between counselor and client may serve as an initial barrier to effective counseling of intersex individuals, and (c) Being born a non-intersex person in this society carries with it certain advantages. Scores on the GICCS range from 29 to 203. School counselors who scored above 116 were deemed to be more effective as counselors working with transgender individuals, and school counselors who scored below 116 were deemed to be less effective as counselors working with transgender individuals. Information about the psychometric properties of the SCTIACS, ICCS, and GICCS, along with the characteristics of participants are found in Simons [11] and Simons and Bahr [25].

## Data analysis

We analyzed data that were gathered from participants who were full time employees, indicated working at a particular school level, were at least 18 years of age, and who reported gender, race/ethnicity, and sexual orientation. We conducted analyses with a high level of confidence despite using a nonrandom sample because approximately 92% of the response

data were completed in full by the participants. Data were analyzed using Spearman's correlations, MANOVAs, ANOVAs and post hoc tests. Spearman's correlations were calculated to assess the strength and direction of the monotonic associations between ordinal variables. MANOVAs were run to measure the associations between two or more dependent variables at the continuous level. This was done after confirming the following: (a) there were not any relationships between the observations in each group of the independent variables or between the groups themselves, (b) outliers were not related to data entry or measurement error, (c) multivariate normality was met and multicollinearity was not present, (d) linear relationships existed between the dependent variables for each group of the independent variables, and (e) homogeneity of variance-covariance matrices and variances were present. If the last assumption was not met, follow-up ANOVAs were conducted.

## Results

To test hypotheses one to three, bivariate correlation analyses were utilized to assess the relationships between ordinal variables and levels of school counselor gender minority advocacy competence: attitudes, gender identity counselor competence, and intersex counselor competence. Spearman's bivariate correlations were calculated, and significant strong positive relationships were found between school counselor gender minority advocacy competence and (a) attitudes toward gender minority students, (b) gender identity counselor competence, and (c) intersex counselor competence. To further test hypothesis three, MANOVAs, ANOVAs, and post hoc tests were used to examine the relationships between school counselor gender minority advocacy competence and age, race/ethnicity, gender, sexual orientation, and school level. Race/ethnicity, gender, sexual orientation, and school level were significantly related to gender minority advocacy competence (see Table 1). However, no relationship was found to exist between age and gender minority advocacy competence, meaning that age was not related to advocacy for and with gender minority students.

**Table 1. Statistically significant School Counselor Transgender Intersex Advocacy Competence.**

| Variable | $M$ | $SD$ |
|---|---|---|
| Gender | | |
| Female | 3.96[a] | 0.85 |
| Male | 4.03[a] | 0.85 |
| Transgender/Nonbinary | 5.40[b] | 0.55 |
| Race/Ethnicity | | |
| African American | 3.56[a] | 0.79 |
| European American | 4.01[b] | 0.86 |
| Multiracial | 4.00[b] | 0.85 |
| Sexual Orientation | | |
| Exclusively Heterosexual | 3.88[a] | 0.84 |
| Mostly Heterosexual | 4.35[b] | 0.75 |
| Exclusively Lesbian/Gay | 4.49[bc] | 0.75 |
| School Level | | |
| Elementary School | 3.69[a] | 0.83 |
| Middle School | 4.06[b] | 0.83 |
| High School | 4.13[b] | 0.82 |

*Note.* TI = transgender and intersex.

Mean values ($M$) that do not have a superscript in common (e.g., x[a] and x[b]) differ significantly from each other at the $p < .0001$ level.

## Analyses for hypothesis testing

Attitudes were found to have a strong positive relationship with school counselor gender minority advocacy competence, meaning that more positive attitudes were related to more competence to advocate for gender minority students (see Table 2). School counselors' levels of gender identity counselor competence were found to have a strong positive relationship with gender minority advocacy competence, meaning that more gender identity counselor competence was related to more competence to advocate for gender minority students. School counselors' levels of intersex counselor competence were found to have a strong positive relationship with gender minority advocacy competence, meaning that more intersex counselor competence was related to more school counselor gender minority advocacy competence. Additionally, a strong significant correlation existed between intersex counselor competence and gender identity counselor competence ($r = .63$). Thus, more intersex counselor competence was related to more gender identity counselor competence.

## MANOVA analysis for age

Responses were assigned to five categories: (a) 20 to 29, (b) 30 to 39, (c) 40 to 49 (d) 50 to 59, and (e) 60 to 69. One-hundred-fifteen school counselor were between 20 and 29 years old, 359 were between 30 and 39 years old, 390 were between 40 and 49 years old, 243 were between 50 and 59 years old, and 84 were between 60 and 69 years old. Significant differences, however, were not found between the five groups of school counselors' mean scores on age, $F(4, 1191) = 1.549$, $p = .186$, $\omega^2 = .482$. These results suggest that age is not related to gender minority advocacy competence, and school counselors can become more effective at helping gender minority students at any age. As a result, counselor educators should teach current and future school counselors about learning from the totality of one's work; use of competency-based assessments can be a part of this education [18, 19].

## MANOVA analysis for gender

Responses were assigned to three categories: (a) male, (b) female, and (c) other comprising transgender respondents. One-hundred-sixty-one school counselors identified as male, 1,025 school counselors identified as female, and five identified as transgender. Significant differences on gender were found, $F(20, 2360) = 4.265$, $p < .0005$; Pillai's Trace = .070; partial $\eta^2 = .035$. Subsequently, ANOVAs were calculated. School counselors who identified as female ($M = 3.96$, $SD = 0.85$) and male ($M = 4.03$, $SD = 0.85$) scored lower on gender minority advocacy competence than school counselors who identified transgender ($M = 5.40$, $SD = 0.55$). These results suggest that gender is related to gender minority advocacy competence; however, these findings of significant differences between members of different gender identity groups based on levels of school counselor gender minority advocacy competence should be interpreted with caution. The sample consisted of 1,191 participants and only five identified as transgender, yet the number of transgender participants outnumbered the number of gender

**Table 2. Correlations between three continuous variables and gender minority advocacy competence.**

| Variable | Gender Minority Advocacy Competence | N | p |
|---|---|---|---|
| Attitudes Toward TI Students | .58** | 1191 | < .001 |
| Gender Identity Counselor Competence | .71** | 1191 | < .001 |
| Intersex Counselor Competence | .54** | 1191 | < .001 |

** $p < .01$.

identity groups. As a result, some would argue that cell size was insufficient to conduct analysis whereas others would not.

## MANOVA analysis for race/ethnicity

Responses were assigned to four categories: (a) African American, (b) European American, (c) Hispanic, and (d) Multiracial. Sixty-four counselors identified as African American, 1,043 identified as European American, 30 identified as Hispanic, and 54 identified as Multiracial. Significant differences were found on race/ethnicity, $F(30, 3540) = 2.613$, $p < .0005$; Pillai's Trace = .065; partial $\eta^2 = .022$. Subsequently, ANOVAs were calculated. School counselors who identified as African American ($M = 3.56$, $SD = 0.79$) scored lower on gender minority advocacy competence than school counselors who identified as European American ($M = 4.01$, $SD = 0.86$) and Multiracial ($M = 4.00$, $SD = 0.85$). However, those in the Hispanic counseling subgroup ($n = 30$) did not differ significantly on school counselor gender minority advocacy competence when compared to other counseling subgroups. Additional research is warranted. These results suggest that race/ethnicity is related to gender minority advocacy competence. School counselors who identified as African American reported lower levels of gender minority advocacy competence than those who identified as European American and Multiracial.

## MANOVA analysis for sexual orientation

Responses were assigned to four categories: (a) exclusively heterosexual, (b) mostly heterosexual, (c) exclusively lesbian and gay, and (d) other. Nine-hundred-ninety-seven school counselors identified as exclusively heterosexual, 99 identified as mostly heterosexual, 47 identified as exclusively lesbian or gay, and 48 identified as other. Significant differences were found on sexual orientation, $F(30, 3540) = 9.633$, $p < .0005$; Pillai's Trace = .226; partial $\eta^2 = .075$. Subsequently, ANOVAs were calculated. School counselors who identified as exclusively heterosexual ($M = 3.88$, $SD = 0.84$) scored lower on gender minority advocacy competence than school counselors who identified as mostly heterosexual ($M = 4.35$, $SD = 0.75$) and exclusively lesbian or gay ($M = 4.49$, $SD = 0.75$). These results suggest that sexual orientation is related to gender minority advocacy competence. School counselors who identified as exclusively heterosexual indicated having lower levels of gender minority advocacy competence than those who identified as either mostly heterosexual or exclusively lesbian or gay.

## MANOVA analysis for school level placement

Responses were assigned to seven categories: (a) kindergarten and elementary school, (b) elementary school, (c) middle school, (d) high school (e) middle and high school, (f) all levels, and (g) other. Thirty-seven school counselors reported working at the kindergarten and elementary school levels, 214 reported working at the elementary school level, 248 reported working at the middle school level, 506 reported working at the high school level, 18 reported working at all school levels, and 130 reported working at a combination of all other levels. Significant differences were found, $F(60, 7080) = 2.597$, $p < .0005$; Pillai's Trace = .129; partial $\eta^2 = .022$. Next, ANOVAs were calculated. School counselors who were working in middle schools ($M = 4.13$, $SD = 0.82$) and high schools ($M = 4.06$, $SD = 0.83$) scored higher on gender minority advocacy competence than school counselors who were working in elementary schools ($M = 3.69$, $SD = 0.83$). These results suggest that school level is related to gender minority advocacy competence. School counselors who worked at the elementary school level did not perceive themselves as being as competent concerning gender minority advocacy as middle and high school counselors.

## Discussion

By having utilized CBAs to measure school counselor gender minority advocacy competence, gender identity counselor competence, and intersex counselor competence in light of IBT and particular demographic variables, we found that attitudes toward gender minority students (transgender and intersex), gender, sexual orientation, and school level placement significantly related to either higher or lower levels of school counselor gender minority advocacy competence. The findings have implications for school counselors, counselor educators, researchers, public policy officials, and others who aim to support gender minority youth. Our findings provide corroborating evidence to support findings from earlier studies. School counselor gender minority advocacy competence was found to relate to positive attitudes, race/ethnicity, and minority sexual orientation [26, 27]. Sexual orientation and gender were also associated with attitudes. School counselors who identified as exclusively gay or lesbian or mostly heterosexual had more favorable attitudes and perceived higher levels of gender minority advocacy competence than school counselors who identified as exclusively heterosexual. School counselors who identified as transgender and nonbinary indicated having the highest levels of gender minority advocacy competence which suggests that these individuals might serve as valuable resources and role models to school counselors who aim to make their school settings more inclusive for gender minorities.

Educators who offer gender minority advocacy training should include elementary school teachers in their training. Training should allow time to complete the SCTIACS self-assessment followed by time to self-reflect over the results. The results may be discussed with others who report having higher levels of gender minority advocacy competence. Thus, while reflection over and discussion of SCTIACS self-assessment results may be easy for some, it may be more difficult for others with lower scores, especially if they have not been exposed to or trained regarding the needs of gender minorities. Research findings suggest that when school counselors try to help themselves and others, they may come to empathize more with students [28]. They may also eventually report higher levels of both intersex and gender identity counselor competence, two areas of counselor competence that were found to positively relate to school counselor gender minority advocacy competence. This suggests that in addition teaching about school counselor gender minority advocacy competence, educators should teach about intersex counselor competence and gender identity counselor competence. This is relevant for two reasons: (a) delivering a comprehensive form of gender minority advocacy is different from delivering counseling services to gender minority students one-on-one, and (b) delivering counseling services to gender minority students, whether within a traditional one-on-one model or a comprehensive form, varies depending on if one is counseling intersex students, transgender students, or another category of gender minority students (e.g., gender non-binary) [1]. For example, unlike transgender students, intersex students may be more likely to undergo a series of genital surgeries. Resultantly, in the future, we hope to see more school counselors and helping professionals in schools self-identify as advocates for gender minority students and come out of the closet if they are also gender minorities. This outcome supports the tenets of IBT and increases the likelihood that more school counselors will outwardly identify as gender minority advocates which increases the likelihood that they will plan and try to actually effectively advocate for gender minority students. In turn, they may be perceived by gender minority students as acting with resilience tied to personhood—proactive, inclusive, and supportive of gender minority students at an individual level and institutional level, about both personal and academic matters. Examples of this can be implicit or explicit and include, for example, talking about being one who identifies as a minority or ally to students; using correct titles, names, and pronouns when trying to motivate and challenge students; and consulting with others who are trustworthy regarding students' needs [11, 29].

## Training implications

Findings inform training implications because school counselors may indicate higher levels of gender minority advocacy competence if they perceive transgender and intersex students favorably, identify as sexual and gender minorities, work with older youth, and indicate higher levels of gender identity and intersex counselor competence. Thus, counselor educators should have school counselors self-administer and complete the GICCS and ICCS, along with the SCTIACS. As aforementioned, scores on the GICCS and ICCS significantly positively correlated with scores on the SCTIACS. Therefore, by teaching school counselors how to provide counseling services to gender minority students effectively at a traditional, one-on-one level, the training may positively affect how school counselors advocate for these students in school settings where they implement comprehensive model (systemic) counseling. Additionally, counselor educators may ask school counselors the following questions: (a) How do you believe your attitudes toward gender minority students influence your willingness to help these students? (b) What knowledge do you need to become more effective at meeting the needs of gender minority students? (c) What skills do you need to become more effective at meeting the needs of gender minority students? (d) How and to what extent are you a personal agent in your position as a school counselor who wants to effectively advocate for gender minority students? (e) How and to what extent are you positively supported as a school counselor who advocates for gender minorities? (f) Which aspects of your personhood as a school counselor relate to your work in this understudied area and why?

In addition to having school counselors self-administer the SCTIACS, GICCS, and ICCS as part of training, counselor educators facilitate discussion after administration and scoring by asking trainees the above questions. This is important because the impact of education might be limited to those who self-administer the scales but do not take time to process the results with others who have more expertise or who identify as gender minorities. Our findings support this outcome and recommendation. Moreover, given that attitudes are also tied to personhood, school counselors should always be prepared to debunk myths commonly associated with gender minority people and topics [6]. For example, some people believe that all transgender people have undergone sex reassignment surgery, which is not true.

Counselor educators share accurate information about gender minority people. Items on the SCTIACS, GICCS, and ICCS that may be reviewed as part of this are: (a) Transgender and intersex students should be accepted completely into our society, (b) Transgender and intersex students should not be allowed to cross-dress at school, (c) Transgender and intersex individuals must choose to live as male or female in order to lead healthy lives, (d) I think that my clients should accept some degree of conformity to traditional gender identities, and (e) It is obvious that a relationship involving an intersex person is not as strong as one involving a non-intersex person. We recommend use of these items in trainings as a starting point for reviewing baseline attitude scores on the SCTIACS, GICCS, and ICCS. Next, counselor educators are encouraged to review (a) counselors' knowledge about transgender and intersex communities and (b) school-based advocacy plans that counselors have in place, if any, to limit the impact of minority stress on health and academic success of gender minorities [1, 11, 30]. Counselor educators should also familiarize themselves with Identity Behavior Theory and the position statements from the World Professional Association for Transgender Health; the American School Counselor Association; and the Society for Sexual, Affectional, Intersex, and Gender Expansive Identities [18, 19].

Counselor educators should teach elementary school counselors about childhood gender identity development [30]. Further, they should inform trainees that models of intersex identity development and models of identity development among transgender people of color are

only in the early stages of development [2]. More theoretical models are needed to inform practices and policies. This is an area for future research. In addition to the need for more theoretical models, there is also a need for more role models who demonstrate an ability to promote greater levels of gender minority advocacy competence in schools and society at large. These role models include school counselors but also other educators who self-identify as advocates and members of gender minority communities [30–33]. Part of training homework should include requesting school counselors to observe and help gender minority individuals and allies, both in- and outside of their school communities. This is done to practice undertaking initiatives to promote more gender-inclusive education.

In this study, school counselors mostly identified as cisgender, heterosexual, White, and female. This demographic is reflective of the national body of school counselors in the United States. Counselor educators, therefore, should recruit a more diverse school counselor workforce that includes sexual and gender minorities, as well as racial and ethnic minorities, and encourage more school counselors to participate in cross cultural events where they interview individuals who are culturally different from themselves. School counselors ask these individuals about their family of origin, career, socioeconomic status, and wisdom. Then, after the interview is conducted the counselors write out, reflect over, and discuss their reactions to the interviewees' responses. Events and best times of the year to interview gender minority individuals include Intersex Awareness Day, Intersex Day of Solidarity, and Transgender Days of Remembrance and Solidarity [11]. School counselors and other helping professionals also may learn about gender minorities by volunteering for the World Professional Organization for Transgender Health, the Intersex Society of North America, El/La Para TransLatinas, Trans Student Educational Resources, the Organisation Intersex International, and interACT Advocates for Intersex Youth [11].

## Limitations

Since correlational data were analyzed, we cannot presume that causation existed between examined variables. We used a nonrandom sample, and the study was conducted online and relied on assistance from presidents of state school counselor associations and state boards of education to recruit participants. Some school counselors, therefore, may not have heard about this study nor were able to access and complete the online survey because they did not have a computer or internet access. Another limitation was that counselors might have participated in the study because they were interested in the topic or they wanted to receive the Amazon gift card. This could have skewed the sample. Last, we did not collect any data on the formal training of school counselors to examine how this training might relate to levels of self-perceived competence in counseling gender minorities. In some jurisdictions, counseling credentials can be granted to individuals in other disciplines such as psychology and social work. Additionally, unlike school level placement, school level training was not assessed as a demographic variable. School level placement, however, was; it was found to significantly correlate with levels of gender minority advocacy competence. Consequently, more research is warranted to see if school level training would significantly correlate with levels of gender minority advocacy competence.

## Future research

Today, some gender minority people report that others, including counseling professionals, interact with them as though they have mental disorders. This is unfortunate, especially for those gender minorities who are not affected by mental disorders. Counselor educators and counselors should teach about this. Counselors educators should also teach about gender nonbinary people. Future lessons and studies might explore to degree to which school counselors

understand the experiences of gender non-binary students. Last, research is needed to refine and strengthen the construct of school counselor gender minority advocacy. This may involve assessing other demographic variables (e.g., advocacy and gender identities) and including new items on training tools that address the needs of gender non-binary students, along with the needs of transgender and intersex students. To do this, gender non-binary people should be interviewed about their experiences with stress, health, and resilience. Their responses will help to develop gender non-binary advocacy competencies which will serve as a resource for educators, helping professionals, researchers, and public policy officials. This is the next chapter of research in education, counseling, and health related to sex, gender, and gender identity.

## Conclusion

The aim of this national survey study conducted considering IBT using CBAs has been to examine gender identity counselor competence, intersex counselor competence, and variables related to school counselor agency to enact gender minority advocacy: attitudes, gender, sexual orientation, and school level placement. All variables but age significantly correlated with school counselor gender minority advocacy competence. As a result, our findings have filled a gap in the literature. We now know more about the makeup of school counselors who are likely to advocate for gender minorities comprising transgender and intersex students who may or may not also identify as gender non-binary. Findings also suggest that more exposure to effective training may help current and future counselors recognize how where they work, along with their backgrounds and personhood, relate (or not relate) to their levels of gender minority advocacy competence. Some counselors choose to participate in additional training to gain more exposure to gender minority topics and communities. It may also be beneficial for them to learn how to promote more gender-inclusive education by participating in events where gender minority people and their allies are present.

Last, both school counselors and counselor educators should stive to raise awareness about events and activities (or lack thereof) in professional organizations to advance the interests of gender minorities. These organizations include, but are not limited to, the American School Counselor Association, the American Counseling Association, the American Psychological Association, the National Association of Social Workers, and the Council on Social Work Education. Our study has extended research findings to mainly the experiences of school counselors [12, 15, 16]. Despite this, however, our understanding of school counselor gender non-binary advocacy competence remains limited. While school counselor advocacy for transgender and intersex students might be similar, this may not be the case regarding advocacy for gender non-binary youth. Therefore, we believe that future researchers should also examine a forms of gender minority advocacy competence that includes gender non-binary advocacy competence. Additionally, researchers should examine how school counselor gender minority advocacy competence may be improved considering Identity Behavior Theory and the influence of other demographic variables that were not assessed as part of this study.

## Supporting information

**S1 File. Demographic form items.**
(DOCX)

## Acknowledgments

The author thanks the participants and recognizes Mercy Simons Lab counseling students Matthew Gallo, Jorge Figuereo, and Melissa Williams for assisting with data collection and

analysis. The Simons Lab is dedicated to scholarship on counseling, students' and educators' experiences, academic and career development, and health and wellness.

## Author Contributions

**Conceptualization:** Jack D. Simons.

**Data curation:** Jack D. Simons.

**Formal analysis:** Jack D. Simons.

**Funding acquisition:** Jack D. Simons.

**Investigation:** Jack D. Simons.

**Methodology:** Jack D. Simons.

**Project administration:** Jack D. Simons.

**Resources:** Jack D. Simons.

**Software:** Jack D. Simons.

**Supervision:** Jack D. Simons.

**Validation:** Jack D. Simons.

**Visualization:** Jack D. Simons.

**Writing – original draft:** Jack D. Simons.

**Writing – review & editing:** Jack D. Simons.

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
