## [Decision Letter · Decision Letter 0]

28 Oct 2020

PONE-D-20-29366

Professional School Counselor Advocacy for Transgender and Intersex Students

PLOS ONE

Dear Dr. Simons,

Thank you for submitting your manuscript to PLOS ONE. After careful consideration, we feel that it has merit but does not fully meet PLOS ONE’s publication criteria as it currently stands. Therefore, we invite you to submit a revised version of the manuscript that addresses the points raised during the review process.

I sent your manuscript to two experts in the field for review, and both saw great promise in the manuscript but also noted a range of concerns that make it unpublishable in its current state. I would like to invite you to carefully address and respond to each of the reviewer's concerns, addressing each in a point-by-point fashion, at which time the manuscript will be re-evaluated for its suitability for publication. Please note that R2's review is included as an attachment.

We look forward to receiving your revised manuscript.

Kind regards,

H. Jonathon Rendina, PhD, MPH

Academic Editor

PLOS ONE

Journal Requirements:

2. Please amend either the title on the online submission form (via Edit Submission) or the title in the manuscript so that they are identical.

Reviewers' comments:

Reviewer's Responses to Questions

**Comments to the Author**

1. Is the manuscript technically sound, and do the data support the conclusions?

Reviewer #1: Yes

Reviewer #2: Partly

2. Has the statistical analysis been performed appropriately and rigorously? 

Reviewer #1: I Don't Know

Reviewer #2: Yes

3. Have the authors made all data underlying the findings in their manuscript fully available?

Reviewer #1: Yes

Reviewer #2: Yes

4. Is the manuscript presented in an intelligible fashion and written in standard English?

Reviewer #1: Yes

Reviewer #2: Yes

5. Review Comments to the Author

Reviewer #1: This research addresses a much needed area for the health and wellbeing of genderqueer students.

Introduction

The section on School Counselor TI Advocacy, the heading title seems misleading. The section seems to be more about the need for more expansive training models to include content related to genderqueer experiences.

The introduction as a whole is focused on trans and intersex youth however, the rest of the manuscript the lack of genderqueer informed content in training models seems to be highlighted. I think it may be pertinent to the rationale of the study if the intro noted this limitation upfront.

For part of the introduction, it appears to be a critique on specific studies rather than what does the current body of research tell the field of school counselors regarding the need for trans and intersex (and genderqueer) related training.

The need for CBAs is predicated on Laurie (2011). Has there been more work in this area? Is there a counterargument to using CBAs or are CBAs considered to be best practices?

TPB is widely known, but IBT is not. I think the reader would benefit from a logic model – this would help the reader visually see the connection between IBT and the goal of the current research.

Methods

In the instruments, how were the measures calculated? Mean scores? Sum? Is there a cut off sore in which someone is deemed “competent”?

Results

Did you collect any data on the formal training of school counselors? Are trained psychologists and social workers more “competent” than professional with school counseling degrees? In some major jurisdictions, counseling credentials can be granted to other disciplines like those previously mentioned. Otherwise this needs to be addressed in the limitations.

In the table “Statistically Significant School Counselor Transgender Intersex Advocacy Competence” the p values seem to be buried at the bottom of the table. Given that the focus is the “Statistically Significant” the reader should see p values in its full form; with a similar structure on the subsequent table.

The demographics table doesn’t seem to match the analyses plan and the subsequent MANOVAs

You mention there was an “other” category in the sexual orientation question. Its missing from the table, was the sample size 0?

What was the sample size of Latinx/Hispanic counselors? It also is missing from the table, but its mentioned in the MANOVA section. If none, this should be addressed in the limitations

Discussion

The authors call on TIGQ professional organizations, but that appears this may put the burden on those in need. More attention should be placed on activities (or lack thereof) of the professional counseling organizations like the ASCA, even the APA/NASW/CSWE, in advancing equity among TIGQ student and populations broadly.

I know the ASCA relies heavily on CBAs, but should the field move above and beyond competencies as a training tool and also integrate a live long learning component? These newer models are more ubiquitous with the goal of advancing cultural competency paradigms. Should this also apply to TIGQ-sensitive learning? This may warrant some attention.

A note of consideration: This paper has a potential not to only address TIGQ competencies but also advance the field into a more holistic and reflexive approach.

Reviewer #2: My review is included as an attachment.

The article reports on survey data with a large sample size regarding an important topic. However, there are a number of places where the presentation of information needs clarifying prior to publication. Here are some suggestions: Please see attachment.

6. PLOS authors have the option to publish the peer review history of their article (what does this mean?). If published, this will include your full peer review and any attached files.

Reviewer #1: No

Reviewer #2: **Yes: **Sarah-Jane (SJ) Dodd

---

## [Author Response · Author response to Decision Letter 0]

8 Dec 2020

Mercy College

Social and Behavioral Sciences

Counseling

555 Broadway

Dobbs Ferry, NY 10522

H. Jonathon Rendina, PhD, MPH

Academic Editor

Public Library of Science (PLOS) ONE

1160 Battery St. 

Koshland Building East, Suite 225 

San Francisco, CA 94111 USA

December 4, 2020

Dear Dr. Rendina:

I am pleased to re-submit the manuscript entitled, “School Counselor Advocacy for Gender Minority Students” for potential publication in the PLOS ONE special issue on health and health care in gender diverse communities. The original manuscript was submitted to Drs. Sevelius, Scheim, and Radix. The aim of this study considering Identity Behavioral Theory was to assess the school counselor role in advocating for gender minority students (transgender and intersex students) using competency-based assessments and a demographic form. 

School counselor participants (N = 1,191) completed the School Counselor Transgender Intersex Advocacy Competence Scale, the Intersex Counselor Competence Scale, the Gender Identity Counselor Competence Scale, and a demographic form. Attitudes were found to have a strong positive relationship with gender minority competence. School counselors’ levels of gender minority competence had large positive relationships with their ability to provide individual counseling services to intersex and transgender people. Black identified participants reported lower levels of gender minority competence than those who identified as White and Multiracial. While school counselors who identified as transgender reported the highest levels of gender minority competence, school counselors who identified as exclusively heterosexual had the lowest levels. Elementary school counselors did not perceive themselves as competent as middle and high school counselors. Age of participants was non-significant. This latter finding may suggest that school counselors can become more effective at advocating for gender minority students at any age. As a result, counselor educators should teach school counselors about learning from the totality of one’s training and work experiences throughout life. Use of competency-based assessments can be a part of this process too. This manuscript still presents a study on school counselor gender minority advocacy competence using a national sample of school counselors. I have addressed the requested edits below:

Edit / Comment

The section on School Counselor TI Advocacy, the heading title seems misleading. The section seems to be more about the need for more expansive training models to include content related to genderqueer experiences. / Added the following sentence at end of paragraph: A need exists for more expansive training models to include content related to the experiences of gender minorities, which we now further explore.

Deleted School Counselor TI Advocacy heading

The introduction is focused on trans and intersex youth however, the rest of the manuscript the lack of genderqueer informed content in training models seems to be highlighted. I think it may be pertinent to the rationale of the study if the intro noted this limitation upfront. / Added the following sentence at end of paragraph - Note: Transgender and intersex students are the focus of this study, not genderqueer students, another student population that warrants being studied. 

For part of the introduction, it appears to be a critique on specific studies rather than what does the current body of research tell the field of school counselors regarding the need for trans and intersex (and genderqueer) related training. / Rewrote entire section.

The need for CBAs is predicated on Laurie (2011). Has there been more work in this area? Is there a counterargument to using CBAs or are CBAs considered to be best practices? / Added the following and a new reference. According to Lurie (12), CBAs illustrate models of competency that can be developed and refined by scholars and public policy officials. Educators in the health professions and others who supervise trainees (e.g., counselor educators) use CBAs to gather empirical data to examine and improve best practices. This practice remains widespread despite the minimally held belief by scholars that CBAs overemphasize the development of individual skills to the detriment of not being open to gain knowledge from the totality of one’s learning experience. Brightwell and Grant (13) argued that this outcome weakens the role of trainees, hurts a profession, and puts individuals who receive services from trainees at risk. 

TPB is widely known, but IBT is not. I think the reader would benefit from a logic model – this would help the reader visually see the connection between IBT and the goal of the current research. / Two figures added and rewrote sections to clarify.

In the instruments, how were the measures calculated? Mean scores? Sum? Is there a cut off sore in which someone is deemed “competent”? / Expanded the information to address.

Did you collect any data on the formal training of school counselors? Are trained psychologists and social workers more “competent” than professional with school counseling degrees? In some major jurisdictions, counseling credentials can be granted to other disciplines like those previously mentioned. Otherwise this needs to be addressed in the limitations. Added to limitation section.

In the table “Statistically Significant School Counselor Transgender Intersex Advocacy Competence” the p values seem to be buried at the bottom of the table. Given that the focus is the “Statistically Significant” the reader should see p values in its full form; with a similar structure on the subsequent table. / Looked up p values; all p values were < .001.

As such no new column was added but the footnote was edited to read: Mean values (M) that do not have a superscript in common (e.g., xa and xb) differ significantly from each other at the p < .0001 level.

The demographics table doesn’t seem to match the analyses plan and the subsequent MANOVAs / This was edited to match more correctly.

You mention there was an “other” category in the sexual orientation question. Its missing from the table, was the sample size 0? / Sample size for Other was 48 and a significant correlation was not found so it was not listed in the table.

What was the sample size of Latinx/Hispanic counselors? It also is missing from the table, but its mentioned in the MANOVA section. If none, this should be addressed in the limitations / Thirty people self-identified as Hispanic in the participant sample, thus data analysis was possible. However, this counseling subgroup did not differ significantly on school counselor transgender intersex advocacy competence when compared to other counseling subgroups who were represented in the sample. Additional research in this area is warranted. 

Addressed in findings on pg 19. 

The authors call on TIGQ professional organizations, but that appears this may put the burden on those in need. More attention should be placed on activities (or lack thereof) of the professional counseling organizations like the ASCA, even the APA/NASW/CSWE, in advancing equity among TIGQ student and populations broadly. / Added the following to page 26: They should raise more awareness about the activities (or lack thereof) of professional counseling organizations like the American School Counselor Association and even the American Counseling Association, the American Psychological Association, the National Association of Social Workers, and the Council on Social Work Education, in advancing equity among gender minority student and populations broadly.

I know the ASCA relies heavily on CBAs, but should the field move above and beyond competencies as a training tool and also integrate a live long learning component? These newer models are more ubiquitous with the goal of advancing cultural competency paradigms. Should this also apply to TIGQ-sensitive learning? This may warrant some attention. / Expanded upon on pg. 7; added:

One way, however, to address this concern might be to emphasize the importance of lifelong learning with trainees early in their training. According to Bajis, Chaar, and Moles (14), the inclusion of lifelong learning into CBAs recognizes that competencies change over time due to ethical, social, clinical, and technological considerations, and it is an area that should be further explored. Addressing lifelong learning in CBAs is a newer approach that has been proposed; Lifelong learning in CBAs has the more ubiquitous goal of advancing cultural competency paradigms in a dynamic world that recognizes humanity not just in work like but in society throughout life.(14)

A note of consideration: This paper has a potential not to only address TIGQ competencies but also advance the field into a more holistic and reflexive approach. See above.

Fully explain the notion of transgender intersex advocacy competency. Describe the components of “advocacy competency”? / Rewrote paragraph 1 to clarify this.

Also, outline why you are conflating transgender and intersex in the research and the scale. It is possible to be a transgender advocate but not an intersex one, or to have intersex competency but not transgender competency. / The areas are not conflated herein:

Added to the following to p. 3: As result, the aim of this quantitative study was to assess the school counselor role in advocating for the needs of transgender and intersex students together. This is because although the needs of these students are different, they are more alike than dissimilar. A need also exists to identify more expansive training models to include content related to the experiences of gender minorities. 

Also, at the very end you begin to discuss the omission of gender queer from your survey. If you want to name that genderqueer was not included in your survey or scales then it might be better if this omission is explained and addressed at the beginning of the paper. / Added the following on pp. 3-4: Note: Transgender and intersex students are the focus of this study, not gender non-binary students, another student population that warrants being studied. The decision was made to not include the assessment of school counselor advocacy for these students because although the issues that they face are similar to TI students, they are also different. According to Simons et al. (1), gender non-binary people subscribe to a non-dichotomous gender identity. That is, they claim a gender identity that is not either male or female, and it may be either fluid or stationary.(25,26) While some have an easy time understanding this, others do not.

The omission of intersex from your demographic form should also be mentioned (for example, address what impact, if any, leaving gender open-ended had on people’s comfort or discomfort in disclosing an intersex identity?). / Participants were asked to identity if they possessed an intersex identity. Updated demo form.

All scales utilized in the study should be more clearly explained. Who developed them, where have they been used, how many questions, utilizing what kinds of response scales, and what are their psychometric properties? / More information has been added and the reader has been directed other resources to learn more about the measures beyond the scope of this paper. See pp. 12-13.

The constant use of acronyms for numerous concepts including TI, CBAs, SCTIACS, GICCS, ICCS, IBT, TPB is distracting for the reader. It may be appropriate to use full names in places, especially for the theories. / Spelled out some in the first para of theories on p. 9

In the review of literature, the author mentions 3 studies but no context is provided for these studies. The studies refer to the school setting but no data is provided re the hostile environment encountered in schools and the theoretical foundation for that hostility (e.g. heteronormativity and cisnormativity). In addition, more details about the studies, for example sample size, would be useful to assess their credibility. / Sample sizes added

It would be helpful to define and describe TPB. It would also be helpful to discuss whether IBT was developed in relation to TBP or simply as an alternative, in either case it would be good to know what the limitations were with TPB. / Expanded upon para 1 on p. 9

It would be helpful to discuss the source of the sample, the mailing lists, etc. under the “Participants and Methods” section. Also, discuss whether responses were geographically split between rural and urban, east coast, west coast, or middle of the country, etc. / This information is found in other articles that have been written using this data set. Citations added.

There is reference to imputing missing data and that “most of the data were missing completely at random”, but the amount of missing data is important to know. It is also important to know which data was missing “based on sexual orientation”. / This information has been added. 

When reporting results there are multiple references to “Large positive relationships” for the correlation data. I believe these should be reported based on strength rather than size, so as “strong” rather than “large”. / Edited to reflect this.

The sample consists of 1190 participants and only 5 identified as transgender. I have serious concerns as to whether that cell size is sufficient to determine statistical significance without encountering error. / Added: These results suggest that gender is related to TI advocacy competence; however, they should be interpreted with caution. The sample consisted of 1191 participants and only five identified as transgender, yet the number of transgender participants outnumbered the number of gender identity groups. As a result, some would argue that cell size was sufficient to examine if significant differences existed between members of the gender identity groups.

what are the components of TI competence and to discuss specifically what skills are needed and what resources are available. / This information is found under test descriptions.

Added more info on pp 12-13:

Examples of knowledge assessed include knowledge of stereotypes, the coming out process, transgender identity developmental, and how to assist with gender transition. Examples of skills assessed include promoting anti-bullying policies, providing training and awareness programs, consulting with parents and guardians, and evaluating the strengths and weaknesses school counseling programs.

On page 17 line 366 the author suggests that since the majority of respondents were cisgender, white, females they should be the target of efforts to develop training efforts. 

Such a statement further marginalizes the voices of BIPOC. Instead, I would interrogate your sampling strategies and geographic distribution of the survey to see why your sample skewed towards this demographic, and whether it is reflective of the national body. / This demographic is reflective of the national body. / Rewrote this para on pg. 24: In this study, most school counselors were mostly cisgender, White, females. This demographic is reflective of the national body. Trainers, therefore, should assist with recruiting a more diverse school counselor workforce and encourage more school counselors to participate in events such as Intersex Awareness Day, Intersex Day of Solidarity, and Transgender Days of Remembrance and Solidarity.(3) School counselors may also learn more about TI people by doing volunteer work for the World Professional Organization for Transgender Health, the Intersex Society of North America, El/La Para TransLatinas, Trans Student Educational Resources, the Organisation Intersex International, and interACT Advocates for Intersex Youth. 

An important topic for study and a large (though skewed) sample. / This limitation addressed on p. 25

I approve this publication as a sole author, and the publication has also been approved by responsible authorities at the institution where I presently work. I recognize that the publisher will not be held legally responsible for any claims for compensation. The manuscript has 7,343 words and is 23 pages in length (less the title, abstract, references, acknowledgments, and supporting information). The work described has not been submitted nor published elsewhere. It is original, and it does not duplicate any work, including my own. The manuscript does not contain anything that is abusive, defamatory, libelous, obscene, fraudulent, or illegal. I do not have any conflicts of interest. The manuscript adheres to APA format style and the Code of Ethics (sixth edition). All identifying information in the main anonymous (blind) file has been removed. 

If you have any questions, please contact me at jsimons1@mercy.edu or by phone at +1-212-810-0257. 

Yours truly,

Jack Simons, Ph.D.

---

## [Decision Letter · Decision Letter 1]

14 Jan 2021

PONE-D-20-29366R1

School Counselor Advocacy for Gender Minority Students

PLOS ONE

Dear Dr. Simons,

Thank you for submitting your manuscript to PLOS ONE. After careful consideration, we feel that it has merit but does not fully meet PLOS ONE’s publication criteria as it currently stands. Therefore, we invite you to submit a revised version of the manuscript that addresses the points raised during the review process.

I have sent your revised manuscript to both of the original reviewers who agreed that you have been largely responsive to prior concerns and that the paper’s suitability for publication has been enhanced. At the same time, one reviewer noted a few additional, minor concerns that I would like to invite you to address in a resubmission. At that time, I will provide a final evaluation of the paper and render a final decision.

We look forward to receiving your revised manuscript.

Kind regards,

H. Jonathon Rendina, PhD, MPH

Academic Editor

PLOS ONE

Reviewers' comments:

Reviewer's Responses to Questions

**Comments to the Author**

1. If the authors have adequately addressed your comments raised in a previous round of review and you feel that this manuscript is now acceptable for publication, you may indicate that here to bypass the “Comments to the Author” section, enter your conflict of interest statement in the “Confidential to Editor” section, and submit your "Accept" recommendation.

Reviewer #1: All comments have been addressed

Reviewer #2: (No Response)

2. Is the manuscript technically sound, and do the data support the conclusions?

Reviewer #1: Yes

Reviewer #2: Yes

3. Has the statistical analysis been performed appropriately and rigorously? 

Reviewer #1: Yes

Reviewer #2: Yes

4. Have the authors made all data underlying the findings in their manuscript fully available?

Reviewer #1: Yes

Reviewer #2: Yes

5. Is the manuscript presented in an intelligible fashion and written in standard English?

Reviewer #1: Yes

Reviewer #2: Yes

6. Review Comments to the Author

Reviewer #1: The manuscript addresses a much needed training area in the school based context. The author substantially revised the manuscript based on reviewer comments.

Reviewer #2: The author has addressed the majority of the comments from the prior review. I would recommend the following minor edits prior to publication.

1. The first paragraph still does not explain what makes the transgender and intersex experiences "more similar" - expanding here would help when you make a statement about there being some differences (again not specified) between the two groups on line 73. I would focus on any important distinctions that readers should be aware of as they review your findings.

2. Also, in the introduction, line 58 the definition of nonbinary is out of place here and does not relate to the subsequent sentence even though that starts with "Additionally, ..." (line 60). I would move the defining up and add a definition for trans and intersex, which are not yet defined.

3. Page 5 line 100 the contents of the parentheses are incomplete.

7. PLOS authors have the option to publish the peer review history of their article (what does this mean?). If published, this will include your full peer review and any attached files.

Reviewer #1: No

Reviewer #2: No

---

## [Author Response · Author response to Decision Letter 1]

5 Feb 2021

I have addressed the requested the edits below:

Edit Comment 

The first paragraph still does not explain what makes the transgender and intersex experiences "more similar" – 

expanding here would help when you make a statement about there being some differences (again not specified) between the two groups on line 73. 

I would focus on any important distinctions that readers should be aware of as they review your findings. 

Response

Added more information at the beginning of paragraph 3 to explain similarities and difference regarding transgender vs. intersex experiences: “Examining counselor advocacy for transgender and intersex youth is warranted because the needs of these youth appear to overlap.(2,3) The two groups experience a wide array of feelings; face challenges (e.g., being bullied); and cope with identity development.(2, 3) This is also the case for gender non-binary students.”

I also edited point 1 to make it more clear and related to findings: “We do not know how counselor competence to provide services to transgender students relates to counselor competence to provide services to intersex students. A review of the research literature indicates that no studies have empirically examined these advocacy areas together. This study appears to be the first to do so.”

Edit Comment

Also, in the introduction, line 58 the definition of nonbinary is out of place here and does not relate to the subsequent sentence even though that starts with "Additionally, ..." (line 60). I would move the defining up and add a definition for trans and intersex, which are not yet defined. 

Response

I moved text up from the prior manuscript to form a new paragraph one. Then, I define each of the gender minority groups in paragraph two. New text in paragraph two reads: “In this study gender minority youth comprise those who identify as transgender and intersex. Scholars estimate 150,000 adolescents in the United States identify as transgender.(4) Transgender youth experience incongruent feelings between birth sex and gender identify.(5) Intersex individuals account for one to two percent of the population.(6) Intersex youth possess a normal variation in hormone levels or chromosomes and may experience differences in body characteristics.(7) Some will undergo genital surgery.(8) Some transgender and intersex youth also identify as gender non-binary. Gender non-binary youth, also referred to as genderqueer youth, possess non-dichotomous gender identities that are neither male nor female; the identities of gender non-binary youth may be fluid or fixed, or exist somewhere between female and male (e.g., neutral).(9,10)”

Edit Comment

Page 5 line 100 the contents of the parentheses are incomplete. 

Response

I addressed this edit, and it now reads as follows: “For example, regarding the latter, school counselors should call for and run gender minority and ally groups and promote policies to assist gender minority students to transfer schools if needed (e.g., due to ongoing physical assault).(2)”

Additional edits/notes 

1. IBT figure has been updated as agency variable is now referred to as resilience in the model.

2. Additional references added to define transgender, intersex, and gender non-binary. As result, numbers in the reference list used for citation updated throughout paper and in reference list.

3. A preprint is now available for reference 21.

 21. Simons JD, Bahr MW, Ramdas M. Counselor Competence Gender Identity Scale: Measuring Clinical Bias, Knowledge, and Skills [Internet]. PsyArXiv; 2021. Available from: https://psyarxiv.com/d2mnu

4. Reference 29 is in press with doi number forthcoming.

 29. Simons JD, Russell ST. Educator Interaction with Sexual Minority Youth [Internet]. EdArXiv; 2020. Available from: edarxiv.org/76fa4

---

## [Editor Report · Decision Letter 2]

18 Feb 2021

School Counselor Advocacy for Gender Minority Students

PONE-D-20-29366R2

Dear Dr. Simons,

We’re pleased to inform you that your manuscript has been judged scientifically suitable for publication and will be formally accepted for publication once it meets all outstanding technical requirements.

Kind regards,

H. Jonathon Rendina, PhD, MPH

Academic Editor

PLOS ONE
---

## [Editor Report · Acceptance letter]

23 Feb 2021

PONE-D-20-29366R2 

School counselor advocacy for gender minority students 

Dear Dr. Simons:

I'm pleased to inform you that your manuscript has been deemed suitable for publication in PLOS ONE. Congratulations! Your manuscript is now with our production department. 

Kind regards, 

on behalf of

Dr. H. Jonathon Rendina 

Academic Editor

PLOS ONE